# Use of Standardized and Non-Standardized Tools for Measuring the Risk of Falls and Independence in Clinical Practice

**DOI:** 10.3390/ijerph18063226

**Published:** 2021-03-20

**Authors:** Jan Neugebauer, Valérie Tóthová, Jitka Doležalová

**Affiliations:** Institute of Nursing Midwifery and Social Sciences, Faculty of Health and Social Sciences, University of South Bohemia in Ceske Budejovice, 370 11 Ceske Budejovice, Czech Republic; Tothova@zsf.jcu.cz (V.T.); Dolezj08@zsf.jcu.cz (J.D.)

**Keywords:** standardized tools, physical disabilities, nursing assessment, nursing practice, nurse opinions

## Abstract

(1) Background: The use of standardized tools is regarded as the basis for an evidence-based assessment. The tools enable monitoring of complex events and the effectiveness of adopted interventions. Some healthcare facilities use standardized tools such as the Morse Fall Scale, but many use non-standardized tools created based on patient needs. Our study question was, why are non-standardized tools used when standardized tools are more beneficial and can be statistically evaluated and compared to other results; (2) Methods: We used a quantitative, non-standardized questionnaire to survey 1200 nurses, which was representative sample for the entire Czech Republic. All questionnaires were assessed in two phases (a) the frequency evaluation and descriptive analysis, and (b) hypotheses testing and correlation analyses; (3) Results: We found that the Conley Scale, Barthel test, and IADL test were preferred by many nurses. Furthermore, we found that nurses using standardized assessment scales noticed risk factors significantly more frequently but regarded the increased complexity of care to be psychologically demanding. (4) Conclusions: In patients with physical disabilities, both types of tools (internal non-standardized and standardized) are used to assess the risk of falls and independence; nurses generally welcomed the increase use of standardized tools in their facilities.

## 1. Introduction

Currently, nursing is undergoing rapid development, and its modernization requires identifying specific patient needs and assessing the patient’s status using standardized and non-standardized assessment tools [1]. The use of standardized tools is the basis for an evidence-based assessment [2]. Using this method, both needs and interventions can be monitored, enabling feedback relative to the care provided [1,2,3]. Optimal nursing should include an assessment of patient needs using standardized tools, identification of complex events, implementation of the best interventions, and an evaluation of the effectiveness of the interventions [4,5]. There are many reasons why healthcare facilities do not use standardized tools, such as inadequate technical support, poor utilization of assessment results, inadequate staff training, or negative attitudes among staff [2,4,6]. Nurses often prefer non-standardized tools created based on specific needs, e.g., geriatric patients [7,8]

The assessment of disabled patients has specific features in clinical practice [9]. Continuous nursing assessments enables monitoring mobility issues that are frequently related to independence problems, falls, pressure sores, incontinence, and spasticity [10,11]. Standardized tools, such as the Conley Scale [12], the Morse Fall Scale [13], or the Tinetti Test [14], can be used to assess fall risk. The Barthel Test, a test of Activities of Daily Living (ADL) and the Instrumental Activities of Daily Living (IADL) test can be used to assess independence [15]. Spasticity and incontinence can be controlled through permanent urinary catheterization or drugs reducing muscular tonus [16]. However, the same results can often be achieved using other methods and by keeping detailed nursing records, although the process needs to be continuously monitored [17,18]. In Czech clinical practice, one of the preferred standardized assessment tools is the Braden Scale, which corresponds with nurses’ competencies [9,19]. Many international tools have been constructed for patients with physical disabilities to help nursing staff understand the nature of decreased functionality and, as a result, better determine appropriate interventions [20]. Examples include morning self-care, which requires movement of both the upper and lower extremities as well as an appropriate level of body coordination; additionally, the sensory functions of sight and hearing and preserved perception are needed [21]. Overall, the nursing care of disabled patients is more demanding, both psychologically and physically, and the introduction of standardized assessment tools makes also the nursing care more time-consuming [22]. Generally, these problems can be managed by increasing the awareness of nurses regarding physical disabilities, patient rights, required competencies, and practical training [16,23]. Increased knowledge can also contribute to improved feelings of safety and a decreased fear of interactions with patients [24].

## 2. Materials and Methods

A quantitative questionnaire created by authors was chosen to test three hypotheses and collect other data about the use of selected assessment scales and examine if nurses would welcome the use of these scales in their practice.

### 2.1. Aims and Hypotheses

Clinical nursing in the Czech Republic predominantly uses non-standardized tools, which are not as effective as the various standardized tools used elsewhere in the world. This study is focused on monitoring the clinical use of selected assessment scales of independence and fall risk in patients with physical disabilities. We also focused on the opinion of nurses regarding selected assessment scales and if they would welcome the introduction of any of scales at their facility. Additionally, the physical, psychological, and time-consuming demands associated with the care of patients with physical disabilities were studied. Another goal of the study was the analysis of significant correlations and searching for complicated moments to enable possible solutions.

We also tested three hypotheses: (1) Nurses who use standardized assessment scales would prefer full introduction into clinical practice; (2) Nurses who use standardized assessment scales monitor the potential risks more than nurses who do not use them, and (3) Nurses prefer standardized tools over non-standardized tools.

### 2.2. Sample

The surveyed included clinical nurses, ward nurses, and head nurses. A sample of 1200 nurses, which was representative of the entire Czech Republic, completed the survey questionnaire. The questionnaires were distributed to all regions of the country and across all types of healthcare facilities and nurses (Table 1). A quota selection method was chosen. The number of respondents in each region was the main quota criterion. The sample was created based on statistical data from the Czech Health Statistics Yearbook [25].

Nurses participating in the study had to meet the following criteria: (1) employed by a healthcare or social healthcare facility in one of the above-mentioned positions, (2) have practical experience working with patients with physical disabilities, and (3) work in an inpatient department.

Nurses were recruited through the research departments of each healthcare facility, which confirmed the selection criteria. From 15 to 20 of the biggest healthcare facilities in each region of the Czech Republic were surveyed during two selected periods (December 2018 and January 2019). Nurses were recruited continuously until the desired sample size was achieved. We did not use any other follow-up strategies due to the random nature of the quota selection process.

We focused on all types of nurses, but mostly on shift nurses and their direct superiors, i.e., ward nurses. Our study also included head nurses who manage all wards in each department (for example, the standard care ward and intensive care unit of a surgical department), and main/chief nurses who manage the entire nursing staff at each healthcare facility.

### 2.3. Questionnaire

Our non-standardized questionnaire was designed using information regarding the needs and complications of patients with physical disabilities [21]. The questionnaire contained three groups of questions, i.e., (1) demographic data, (2) an assessment of the difficulty associated with (a) providing care to physically disabled patients and (b) the use of specific assessment tools, and (3) the extent to which the tools were actually used in nursing practice.

The basic information supplied to respondents contained the essential characteristics regarding the goals of our study, a description of our work and results, information on anonymity, GDPR (General Data Protection Refulation) instructions for filling out the questionnaire, and information on the investigating team. 

The demographic data contained four questions about gender, education, the region where the respondent was employed, and the respondent’s position (Table A1 in Appendix A). The demandingness of care was assessed using three closed questions. Possible answer options were “yes,” “no,” and “I don’t know” with regard to fall risk analysis. A 0–10 point scale (least demanding to most demanding) was chosen for the analysis of demandingness. The acceptance of the introduction of selected tools into actual practice was assessed using a Likert-type scale —“maximally,“ “very much,“ “average,” “a little,“ “not at all,” and “I don’t know” (Table A2 in Appendix B). 

### 2.4. Data Collection and Analysis

The pilot version of the questionnaire, the evaluation and resulting corrections of the questionnaire were performed between February and April 2019. As a result, two questions were corrected, and the total number of questions was reduced. The main study was performed between April and September 2019. Data analysis and evaluation followed.

We distributed 1490 questionnaires and 80.5% (*n* = 1200) were completed and returned. Copies of the questionnaire were sent to the directors of the selected facilities, which is established protocol for data collection.

The analysis was performed in two parts. In the first part, frequency tables were developed, and the absolute and relative frequencies and mean values (modus, median, mean), dispersion, standard deviation, range, estimation of dispersion and the standard deviation, interval estimation of the expected value, and significance at the 0.05 level were calculated.

The second part included the construction of contingency tables with absolute and relative frequencies. Correlations were analyzed according to the characteristics and number of observations using the Pearson Chi-Square Test and Independence Test. Subsequently, Pearson’s contingency coefficient, Ćuprov’s coefficient, Cramer’s coefficient, the Wallis coefficient, Spearman’s coefficient and correlation coefficient were applied. The weight of individual relationships was measured at three levels of significance—α = 0.05, 0.01, and 0.001.

An analysis of significant correlations was carried out for each variable independently. In the cases with insufficient numbers, the Yates’s correction was applied. In the description of significant correlations, the following notation is used: X^2^—Chi-squared test; p—independence test; df—degrees of freedom; n.s.—non-significant difference; *****—significant difference for the significance level of α = 0.05; ******— significant difference for the significance level of α = 0.01; *******—significant difference for the significance level of α = 0.001.

## 3. Results

### 3.1. Demographic Data

Both women (*n* = 1158; 96.5%) and men (*n* = 42; 3.5%) with a secondary education (*n* = 454; 37.8%), higher professional (*n* = 235; 19.6%), and academic education (*n* = 511; 42.6%) participated in the study. Respondents included nurses working in shifts (*n* = 1144; 95.3%), ward nurses (*n* = 47; 3.9%), head nurses (*n* = 9; 0.7%), and main/chief nurses (*n* = 0; 0%). Statistics and values for each region are presented in Table 1.

### 3.2. Use of Assessment Scales

In Czech nursing, many non-standardized tools, focused on specific problems, are used in each facility. In addition, standardized tools, such as the Conley Scale, MFS (Morse Fall Scale), Tinetti Scale, Barthel Test, and IADL are also used. Nurses in clinical practice prefer the MFS for the assessment of fall risk and the Barthel Test for the assessment of independence (Table 2).

Nurses in clinical practice evaluate the nursing care of patients with physical disabilities to be generally more demanding, subjectively reporting that such patients are physically and psychologically more demanding and require more time-consuming care (Table 3).

A statistically significant correlation was found regarding the use of assessment scales in patients with physical disabilities vs. correctly assessed risk (existence of risk) and the psychological demandingness of the care provided. No statistically significant correlations were found relative to the use of assessment scales in patients with physical disabilities vs. time and physical demandingness (see Table 4).

Nurses who reported the use of assessment scales in their facilities were able to recognize risks significantly more frequently but also reported that the increased complexity of care was psychologically more demanding.

### 3.3. Introduction of Assessment Tools into Nursing Practice

Generally, respondents reported that they would welcome the introduction of selected standardized assessment tools into their facility. Their preferred tools were the Conley Scale for the assessment of fall risk and the IADL to assess independence.

Significant correlations were found relative to the use of assessment tools and the introduction of the Colney Scale, Morse Fall Scale, Tinetti Scale, the ADL, and the IADL (Table 5).

Respondents who use assessment tools were significantly more likely to welcome the introduction of standardized assessment tools for the assessment of fall risk and independence, i.e., the Conley Scale, Morse Fall Scale, Tinetti Scale, ADL, and IADL (Table 6).

## 4. Discussion

### 4.1. Use of Standardized Tools

The use of standardized assessment scales is regarded as very effective since they enable monitoring of patient progress and the creation of statistics that can drive further improvements. Najafpour et al. studied the use of assessment tools focused on individual factors that can decrease the risk of falls. Their study confirmed that the use of standardized tools decreases the incidence, and they recommended healthcare managers consider introducing these tools into departments where falls were of great concern, such as cancer units [26]. Similar conclusions are reported by Kaya et al., who focused on ways to support assessments in clinical practice. Their results showed that recording risks and partial factors are integral parts of nursing care. In addition, they encourage management to develop tools for their staff or their modifications that are directly focused on sections and are highly practical. Testing tools used by a particular unit should be developed for nurses starting to work at the unit [27].

Virtually all healthcare facilities use scales to assess fall risk, the risk of developing pressure sores, independence, cognitive functions, and physical or mental health. However, the integration of these strategies into individual units can sometimes be very de-motivating for nurses [28]. This study draws attention to the introduction of specific tools or modifications of tools without consulting clinical nurses regarding optimal methods for implementation. As a result, nurses routinely complete the assessment tool without thinking since it is “required documentation.” Therefore, the results of the assessment are not properly integrated into the patient’s care plan. This problem was mentioned by Achrekar et al. Their study pointed out that many patient records were incomplete due to the assessment being routine and sometimes even pointless. The inclusion of standardized tools into patient care is often regarded as time-consuming, and the need to later deal with the results makes the work more psychologically demanding [29].

### 4.2. Independence Assessment

According to Pashmdarfard and Azad, ADL and IADL can be used to assess independence. The basic versions of the ADL and IADL (which is recommended for use in all healthcare facilities) are regarded as widespread and popular tools. Pashmdarfard and Azad also supports the results of our study, which demonstrates the usability of these standardized tools in clinical practice [15]. On the other hand, Osakwe et al. disagrees with these observations and regard the original versions of the ADL and IADL to be antiquated and recommend a more modern tool that focuses more on domains and factors enabling a better assessment of independence for each patient [30]. Liebzeit et al. also supports this opinion. They regard the Katz Index or other specifically focused tool as better options for the assessment of patients with physical disabilities. These tools are traditionally used in elder-care homes (nursing homes), and their reliability and validity have been tested many times; no better tool has been found for the assessment of independence [31]. A study by Roedl et al. expresses equally positive opinions regarding independence assessment using the Barthel Index and Katz Index. In elderly individuals, where a physical disability is often present, the Katz Index is generally a better choice. The absence of a mobility assessment makes it a universal tool for healthcare facilities where patients are under the supervision of others [32].

The results of our study draw attention to the overall support for the introduction of these tools into clinical practice. The respondents in our study reported that they would welcome the introduction of the ADL and IADL to their facilities. While Yi et al. recommend assessments using the Barthel Test, they also recommended that the assessment tool be tested using the intended target group prior to introducing it into practice. Their study shows the Barthel Test to be unsuitable for use in patients with dementia, in whom the results of independence tests are almost always unreliable and do not correspond with reality [33].

### 4.3. Risk of Fall Assessment

Fall risk is generally associated with the elderly. Park et al. focused on tools for assessing fall risk. The Tinetti Scale and the Timed Up and Go Test, which is easily done in clinical practice, were found to be suitable at assessing the risk of falls. In patients with a physical disability, the Timed Up and Go Test may not always be the best option. Therefore, the Tinetti Scale or other more specific scales, i.e., more focused on the assessment of risk factors, are likely to be more suitable for the assessment of fall risk [34]. Rivolta et al. are of the same opinion and recommend the Tinetti Scale for clinical practice. In their study, the Tinetti Scale was enhanced by using an instrument placed on the patient’s chest that monitored coordination. This modification can be introduced into physical therapy units and for patients who have no significant gait defects and are not dependent on supportive aids, such as wheelchairs [35].

The Morse Fall Scale is another tool focused on fall risk assessment, and the results of our study support its introduction into clinical practice. Pasa et al. regard the Morse Fall Scale to be a highly valid scale for the assessments of fall risk in most patients, not only elderly patients. Their study tested this tool using continuous measurements in patients from admission until discharge from the unit. The difference increased only in 4.6% of patients [36]. Gringauz et al. also agrees with the results of our study and the results of other authors. Their study focused on a variety of patients, including disabled patients dependent on wheelchairs. The tool was suitable even for those purposes and is very useful in practice [37]. The tool was also supported by Miertova et al., who recommend it for use in emergency care and for patients with neurological diseases [38].

### 4.4. Study Limitations and Recommendations

The limit of this study was the use of a non-standardized survey questionnaire, which was constructed for our study based on the available literature; as such, we have no relevant results for comparison. In some cases, the number of responses to certain questions was too small for analysis. Because of the low representation of men in Czech nursing practice, we were unable to draw any conclusions relative to gender. All research data are relevant for other countries with similar same educational systems and nursing competencies.

We recommend using standardized questionnaires for measuring satisfaction with actual providing care and used assessment tools for measuring people with any type of disability. We also recommend assessing patients with different types of physical disability separately, for example, only patients after spinal cord injury, after amputation, or patients with paraplegia.

For future research, we recommend testing new assessment tools that have been standardized and are widely used around the world. Based on our study, nurses welcomed the introduction of standardized assessment tools, especially if the tools are focused on specific needs and risk factors.

## 5. Conclusions

Nurses caring for patients with physical disabilities find assessments using standardized tools to be helpful and effective. Internal, non-standardized assessment tools (generally created based on patient needs) and standardized tools, such as the ADL and IADL, are commonly used to assess independence, and the Tinetti Scale, Conley Scale, and Morse Fall Scale are used to assess the risk of falls.

Our literature search confirmed that the introduction of standardized assessments is useful, and the Morse Fall Scale, ADL, and IADL are suitable tools, although outdated. For patients with physical disabilities, the Katz Index or other modified tools, which do not focus directly on gait, are better options.

It was shown that nurses who were familiar with standardized tools would welcome their introduction into all clinical practice. They also noted that assessment tools brought fall risks to their attention more frequently, although they found the increased complexity of care to be more demanding. Based on our study, nurses in the Czech Republic prefer using standardized tools. However, managers need to be better acquainted with the types of patients in their facilities and know which assessment tools is best suited for each type of assessment. Since there are many standardized tools available, it is important that the tools are chosen and used in accordance with nurse competencies.

## Figures and Tables

**Table 1 ijerph-18-03226-t001:** Statistical data for each region.

Region	Absolute Frequency	Relative Frequency
Prague, (capital)	215	17.9%
Central Bohemia	103	8.6%
South Bohemia	67	5.6%
Pilsen Region	66	5.5%
Region of Karlovy Vary	34	2.8%
Region of Ústí nad Labem	87	7.2%
Liberec Region	38	3.2%
Region of Hradec Králové	63	5.2%
Pardubice Region	50	4.2%
Highlands (Vysočina)	58	4.8%
South Moravia	148	12.3%
Olomouc Region	81	6.8%
Zlín Region	58	4.8%
Moravian-Silesien Region	132	11.0%
TOTAL	1200	100%

**Table 2 ijerph-18-03226-t002:** Usability of standardized tools in the Czech practice.

Tool	Mean	Modus	Median	Dispersion	Standard Deviation	Interval Estimation of the Expected Value of 0.05	Interval Estimation of Dispersion of 0.05
Conley scale	2.838	3	3	2.203	1.4845	2.838 ± 0.084 2.838 ± 0.084	2.204 − 0.166 2.204 + 0.187
Morse Fall Scale	2.2	1	2	2.647	1.627	2.2 ± 0.092 2.2 ± 0.092	2.647 − 0.02 2.647 + 0.225
Tinetti scale	3.097	3	3	1.844	1.358	3.097 ± 0.077 3.097 ± 0.077	1.844 − 0.139 1.844 + 0.157
Barthel test (ADL)	2.061	1	2	1.844	1.358	3.097 ± 0.077 3.097 ± 0.077	1.844 − 0.139 1.844 + 0.157
IADL	2.172	1	2	2.429	1.559	2.173 ± 0.088 2.173 ± 0.088	2.429 − 0.183 2.429 + 0.207

**Table 3 ijerph-18-03226-t003:** Subjective feelings regarding demandingness (on a 0–10 scale).

	Mean	Modus	Median	Dispersion	Standard Deviation	Interval Estimation of the Expected Value of 0.5	Interval Estimation of Dispersion of 0.05
Time consuming Demandingness	9.54	10	9	1.1269	1.0616	9.538 ± 0.060 9.538 ± 0.060	1.127 − 0.085 1.127 + 0.096
Physical demandingness	9.50	10	9	1.0417	1.0206	9.500 ± 0.058 9.500 ± 0.058	1.042 − 0.079 1.042 + 0.089
Psychological demandingness	9.05	10	10	3.2207	1.7946	9.052 ± 0.102 9.052 ± 0.102	3.221 − 0.243 3.221 + 0.274

**Table 4 ijerph-18-03226-t004:** Correlations between the use of assessment scales and the demandingness of care.

Use of Assessment Tools and	Value x^2^	df	*p*	Significance
Existence of risk	282.618	4	<0.001	***
Time demandingness of nursing care	2.449	2	0.294	n. s.
Physical demandingness of nursing care	4.075	2	0.130	n. s.
Psychological demandingness of nursing care	125.876	2	<0.001	***

**Table 5 ijerph-18-03226-t005:** Introduction of tools into respondents’ facility.

Tool	Mean	Modus	Median	Dispersion	Standard Deviation	Interval Estimation of the Expected Value of 0.05	Interval Estimation of Dispersion of 0.05
Conley scale	2.055	1	2	0.942	0.9706	2.055 ± 0.055 2.055 ± 0.055	0.942 − 0.071 0.942 + 0.080
Morse Fall Scale	2.3925	3	3	0.903	0.9504	2.393 ± 0.054 2.393 ± 0.054	0.903 − 0.068 0.903 + 0.077
Tinetti scale	2.3408	3	3	1.066	1.0672	2.341 ± 0.059 2.340 ± 0.058	1.066 − 0.081 1.066 + 0.091
Barthel test (ADL)	2.385	3	3	0.755	0.869	2.385 ± 0.492 2.385 ± 0.049	0.755 − 0.057 0.755 + 0.0642
IADL	2.107	1	2	1.22	1.103	2.107 ± 0.062 2.107 ± 0.062	1.217 − 0.092 1.217 + 0.104

**Table 6 ijerph-18-03226-t006:** Correlations between the use of tools and their introduction into practice.

Use of Assessment Tools and ….	Value x^2^	df	*p*	Significance
Assessment of fall risks according to Conley	351.662	8	<0.001	***
Assessment of fall risks according to Morse Fall Scale	382.559	8	<0.001	***
Assessment of balance test according to Tinetti	98.622	8	<0.001	***
Assessment of independence—Barthel test	221.484	8	<0.001	***
Assessment using IADL	236.815	8	<0.001	***

## Data Availability

All data are available at the university library and project office of the University of South Bohemia, Ceske Budejovice.

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
