# Peer review of "Use of Standardized and Non-Standardized Tools for Measuring the Risk of Falls and Independence in Clinical Practice"

_ijerph, 2021, doi:10.3390/ijerph18063226_

Round 1
Reviewer 1 Report
Referee:
1.- I have read over the manuscript " Relationship between the Use of Assessment Scales in Patients with Physical Disabilities and the Scales Introduction into the Nursing Practice.
2.- There is an error on line 38, it says "bwewst" and it should say "best".
3.- I believe that the authors should explain in more detail how the sample of nurses participating in the study was constructed, identifying their representativeness with respect to the total population of nurses in the Czech Republic.
Author Response
Dear reviewer,
I very appreciate your recommendations.
Your recommendation: "There is an error on line 38, it says "bwewst" and it should say "best". " has been fixed.
Also, the next one - identifying how we create a representative sample of nurses.
Thank you again and have a great day.
Best regards,
Correspond author
Reviewer 2 Report
Dear authors,
First of all, congratulations on your research paper. It's a core topic for nursing care, particularly regarding the quality of produced outcomes.
I have some comments and suggestions for further improvement of your work.
General considerations:
- Authors should review citations in the text, using numbers in [brackets], instead of names of authors and years. Also, please review the journal's instructions for authors and STROBE statement, in order to help your structure and report your data.
- After the title reading, the reader will think that the paper will generally approach assessment scales for physical disability. But, after abstract reading, the reader finds out that the two main health problems addressed are ADL independence and Falls. I suggest that authors study a way to be more coherent between title and abstract since these elements are very important.
- The paper needs an extensive written English review (some typos: bwewst, line 38).
Abstract:
- The purpose of the study is missing in the (1) Background section.
- According to the IJERPH author instructions, the abstract has a max. word count of 200. The authors present an abstract with 208 words.
- The phrase 'The whole care becomes psychologically and physically more demanding and more time consuming' (Lines 14-15) is somehow out of context with the previous sentences.
- Avoid starting phrases with numbers (eg., '1,200 questionnaires...'; line 17).
- Avoid using 'etc' per system.
Introduction
- It would be necessary to early define, distinguish and give examples of 'standardized assessment tools' and 'non-standardized assessment tools'.
- I suggest writing 'complex events' instead of 'complicated situations'.
- I deem it necessary to include your study's purpose in the Introduction section.
- The authors present a poor and somehow confusing description of the rationale of the investigation being reported. The objectives seem to focus ADL independence and falls, but the Introduction talks about many other concepts (pressure sores, spasticity, etc). Authors should focus on giving a background on assessment tools of their main concepts in the study, and support why it is important to undergo this investigation. Namely, that literature states that using validated instruments to assess fall risk and ADL independence is an advantage to generate positive health outcomes. Nonetheless, in the Czech Republic there is no standard tool, or there is no study that monitors this topic.
- The authors cite a systematic review (Wind et al., 2005). There must be a more updated systematic review.
Materials & Methods
- Using 'non-standardized questionnaires' might mislead the reader when it comes to thinking about 'non-standardized tools'. I suggest using another term.
- The hypotheses presented are not in accordance with the aims/objectives of the study.
- A point for discussion: if nurses use standardized assessment scales, aren't those tools already introduced in clinical practice? What do you mean by 'prefer their introduction into clinical practice'?
- In your 'Aims and Hypotheses' section, I would include that the study approaches the Czech Republic context. This information would also be useful in the Introduction, in the purpose sentence.
- In your 'Sample' selection, it would be recommended to describe how were nurses recruited. Which method was used? Randomized, convenient, and others. Also, recruitment period and follow-up strategies.
- The authors should explain how they arrived at 1,200 nurses.
- I would suggest that you mention Table 1 in the '3.1 Demographic Data'.
- On your 'Questionnaire' section, I deem it necessary to mention whether the data collection instrument was constructed by the authors or adapted from an external source. Also, it would be very recommended to describe the questions you present in Appendix, namely saying that they are Likert-type questions varying from x to y, for example.
- On your 'Data collection and analysis' section, avoid starting phrases with numbers (line 122). Also, authors should describe which are the criteria of data collection and what do they mean by 'first and second degree of classification'.
Results
- The Demographical part of the survey (Appendix 1) mentions 'Main nurses', which are not reported in demographic data on the Results section. There were no 'main nurses'? Also, it would be interesting to briefly describe each nursing category, since they differ between countries.
- On the footer of Table 2, there are two numbers with a description (1. Second item; 2. Third item). What do tey refer to?
Discussion
- Line 237, authors state that 'Their study also supports the results of our study, which shows the usability of these standardized tools for clinical practice'. But, the cited assessment tools, namely Katz Index, FAI, TFL, etc, were not used by the authors.
- Authors should include study limitations and recommendations for future research in this field. Also, if there is any bias in the reported results. It would also be important to mention the external validity of the study, or if the results can only generate conclusions for the Czech context.
Conclusions
- The conclusion doesn't answer the 2nd and 3rd objectives mentioned in the methods.
Appendix B, Table A2: there's a typo in the title ('usable tor Czech pracice?).
References: use abbreviated journal names.
Author Response
Dear reviewer,
thank you so much for exhausting review. We very appreciate your recommendations.
Based on your informations we update title, abstract and other informations. Also we fix the citation issues in all text, now we use a numbers in brackets.
In all document we change the sentences where we started with numbers and fixed "etc".
All our changes has been marked by system in MS Word - tracking changes
Abstract
- We used a shorter sentences and now we have less than 200 word.
- In backround section we try to breafly explain the purpouse of this study.
- We change the incorrect sencence what is not in context with the other sentences and we also change a starting phrases with nubmers
Introduction
- In this section we create examples of standardized and non-standardized assessment tools.
- We changed the "complicated situations" for "complex events"
- The purpose of our study was included to this section
- We fixed a systematic review from 2005 and changed it
Materials & Methods
- We used another term for non-standardized questionaire.
- Our hypothesis are now in accordace with objective
- And we fixed the missinformations about your question. "if nurses use standardized assessment scales, aren't those tools already introduced in clinical practice". - In Czech republic we used many tools for measure patients. But anytime the people use a standardized with non-standardized tools in same time from bad reason - the facility management want it. The nurses wanted to use one of them
- The hypotheses presented are not in accordance with the aims/objectives of the study
- We describe a Czech Republic context in introduction part for better information
- Also, we fill into this section how we recruited the nurses and how we took a 1,200 nurses as a representative sample
- We used better description for used questionnaire.
Results
- In this section the re is an issue with the "main nurses" answer. We took this answer also into the result section.
Discussion
- We fixed a missinformations in this section
- This section now include th estudy limitations and recommendations for future research.
Conclusions
- The conclusion part now correspond with all objectives
Appendix B, Table A2: we fixed the issue.
References: we use abbreviated journal names.
Once more we thank you for your excellent work.
Have a great day and stay healthy.
Best regards,
Corresponding author
Reviewer 3 Report
Introduction: I believe that the current literature should justify the importance of patient assessment and its recording in the medical record.
Conclusions: this section should respond to all the objectives considered in the study.
Author Response
Dear reviewer,
Thank you very much for your recommendations. We appreciate it.
You recommend to update a introduction, so based on this we fix any issues and we fill into the text other informations.
Also you recommend to rework the conclusion section. Based on your opinions we fix this issues and fill into this section informations about all our objectives.
I would like to thank you again for your recommendation
Have a great day and stay healthy.
Best regards,
Corresponding author
Round 2
Reviewer 2 Report
Dear authors,
Congratulations on your work. It was a pleasure to read and review your paper.